# Effects of 2HDM in Electroweak Phase Transition

**Arnab Chaudhuri** [1,*,†] , **Maxim Yu. Khlopov** [2,3,4,†] **and Shiladitya Porey** [1]

1    Department of Physics, Novosibirsk State University, Pirogova 2, 630090 Novosibirsk, Russia; shiladitya@g.nsu.ru
2    CNRS, Astroparticule et Cosmologie, Université de Paris, F-75013 Paris, France; khlopov@apc.in2p3.fr
3    Institute of Physics, Southern Federal University, 344090 Rostov on Don, Russia
4    Center for Cosmopartilce Physics "Cosmion", National Research Nuclear University "MEPhI"
     (Moscow Engineering Physics Institute), 115409 Moscow, Russia
*    Correspondence: arnabchaudhuri.7@gmail.com
†    These authors contributed equally to this work.

**Abstract:** The entropy production scenarios due to the electroweak phase transition (EWPT) in the framework of the minimal extension of standard model, namely the two Higgs doublet model (2HDM), are revisited. The possibility of first order phase transition is discussed. Intense parameter scanning was done with the help of BSMPT, a C++ package. We perform numerical calculations in order to calculate the entropy production with numerous benchmark points.

**Keywords:** electroweak phase transition; 2HDM; extended standard model; entropy





## 1. Introduction

On 7 April 2021, muon g-2 experiments at FERMILAB [1] published the magnetic moment of muons, and the result has drawn attention because the value of the magnetic moment appears to be incongruous/anomalous as it deviates from the theoretical value predicted in the Standard Model (SM). At the tree-level of quantum electrodynamics, the gyromagnetic ratio of a muon is $g_\mu = 2$. Adding virtual particles and loop corrections, the calculated value of $(g_\mu - 2)/2$ in SM was $116{,}591{,}810(43) \times 10^{-11}$, (see Equation (3) and Equation (2) of [2]) while the measured value from FERMILAB was $116{,}592{,}061(41) \times 10^{-11}$ [2,3]. The detection of this anomaly has buttressed previous claims about the disaccord between the theoretical and experimental results [4]. This enduring disagreement may be an indication of the existence of theories beyond the standard model (BSM), e.g., supersymmetric theory [5].

On the other side, the Electroweak Phase Transition (EWPT) in the early hot universe in the SM framework with a single Higgs field is also a well-established theory. According to this theory, when the temperature of the universe drops to near a critical value ($T_c$, for details see [6] or Section 2), another minimum of the Higgs potential for a non-zero value of Higgs fields appears, and the phase transition ($SU(2)_L \times U(1)_Y \to U(1)_{em}$) happens; consequently, the intermediate bosons and fermions gain masses. The parameter-space of this EWPT might be altered if EWPT happens in the BSM model.

Here, we consider the 2HDM model, which provides a minimal phenomenological description of some effects of the supersymmetric model predicting a two Higgs boson doublet. For more motivation of this theory, see [7,8]. Some extensions of 2HDM can not only produce dark matter (DM) but also have the potential to successfully explain the $g_\mu$ anomaly of muons [9]. With the parameter space presented in [10], a strong first order EW phase transition can successfully explain the g-2 anomaly.

In this paper, we considered EWPT in real type-I 2HDM and did not take into account any g-2 anomaly. Hence, the parameter space in the cited paper is beyond the scope of our work. A strong first order EWPT can destroy the thermal equilibrium state of the universe [11]. A first order phase transition is realised when the ratio $VEV/T_c \geq 1$, and this can produce a substantial amount of entropy [12]. In this work, we discuss one of the

thermodynamical properties, namely entropy production during EWPT by scanning the parameter space in detail for Type-I real 2HDM scenarios.

At a very primeval time, when the temperature of the universe $T(t) \gg T_c$, the universe was dominated by ultra-relativistic species and $\Gamma / \mathcal{H}(t) \sim T(t)^{-1} \ll 1$, where $\Gamma$ is the rate of particle interaction with each other and with the photons and $\mathcal{H}(t)$ is the Hubble parameter. All the species, therefore, were in thermal equilibrium with photons. During that aeon, almost all of the fermions and bosons were massless, and the contribution from those who were already massive (specifically speaking, decoupled components, e.g., decoupled DM particles, like axions) to the total energy density of the universe was insignificant.

During that time, $T \gg \left( \mu_{\text{chm. pot.}} - m \right)$, $\mu_{\text{chm. pot.}}$ represents the chemical potential of a particle [13], and thus massless bosons had zero chemical potential, and we assume the chemical potential of the fermions to be negligible. The entropy density per comoving volume $s(t)$ was conserved and is expressed as

$$s(t) \equiv \frac{\rho_r(t) + P_r(t)}{T(t)} a(t)^3 = \text{const.} \tag{1}$$

where $\rho_r(t)$ and $P_r(t)$ are the energy and pressure density of the relativistic components, respectively, and $a(t)$ is the scale factor at that moment. For relativistic particles,

$$s(t) \sim g_{\star,s} \, T(t)^3 \tag{2}$$

$g_{\star,s}$ is the effective number of degrees of freedom in entropy, and it is not constant over time. Although it depends on the components of the primordial hot soup, it is not always the same as $g_\star$, which is the effective number of relativistic degrees of freedom (for discussion, see [14]). However, for this project, we assumed $g_{\star,s} \approx g_\star$. Now, from Equations (1) and (2), we obtain $T \sim a^{-1}$. With the expansion and cooling down of the universe, at some epoch, the universe went to the state of thermal inequilibrium and, hence, the value of $s$ and, thus, $g_\star(T) a^3 T^3$, might have increased as entropy can only either increase or remain constant.

During EWPT, as the temperature drops below from $T_c$ up to the decoupled temperature of a particular component of the relativistic plasma, that component decouples and becomes non-relativistic and massive. The decoupled temperature depends on the masses of the components and their respective coupling constants. This decoupling process causes the change in $g_\star$ of the relativistic plasma. Electroweak baryogenesis can happen during EWPT and, hence, thermal inequilibrium is established in the universe following Sakharov's principle [15].

This results in violation of the entropy conservation law, and a net influx of entropy is generated. This led to the increase of $s$. The main contribution to this influx comes from the single heaviest particle with $m(T) < T$. Hence, for this temperature range, $g_\star(T)T = \text{const.}$ and the entropy rises by $a^3 T^3$. Since the final temperature $T = m(T_f)$ (below which a new particle starts to dominate) is not dependent on $g_\star$, we can say that the entropy increased in the background of $g_\star a^3 T^3$.

This paper is arranged as follows: In Section 2, we give the theoretical framework of EWPT in 2HDM. In Section 3, details about the BSMPT package and numerical calculations as well as the results of this work are presented, and this is followed by a generic conclusion in Section 4.

## 2. Theoretical Framework: EWPT Theory in 2HDM

The Lagrangian density of EWPT theory within 2HDM is:

$$\mathcal{L} = \mathcal{L}_f + \mathcal{L}_{\text{Yuk}} + \mathcal{L}_{\text{gauge,kin}} + \mathcal{L}_{\text{Higgs, kin}} - V_{\text{tot}}(\Phi_1, \Phi_2, T) \tag{3}$$

The first four terms on the right hand side of Equation (3) are

$$\mathcal{L}_f = \sum_j i\left(\bar{\Psi}_L^{(j)} \not{D}\Psi_L^{(j)} + \bar{\Psi}_R^{(j)} \not{D}\Psi_R^{(j)}\right) \tag{4}$$

$$\mathcal{L}_{\text{Yuk}} = -\left[y_e \bar{e}_R \Phi_a^\dagger L_L + y_e^* \bar{L}_L \Phi_a^\dagger e_R + \cdots\right] \tag{5}$$

$$\mathcal{L}_{\text{Higgs}} = (D^\mu \Phi_a)^\dagger (D_\mu \Phi_a) \tag{6}$$

$$\mathcal{L}_{\text{gauge,kin}} = -\frac{1}{4}G_{\mu\nu}^j G^{j\mu\nu} - \frac{1}{4}F_{\mu\nu}^B F^{B\mu\nu} \tag{7}$$

where $i = \sqrt{-1}$, $\Psi^{(j)}$ is the Dirac field for the $j$-th fermion species, subscript $L$ ($R$) is for the left (right) chiral field, and subscript $a = 1, 2$ is for the two Higgs fields. In Equation (5), $y_e$ is the Yukawa coupling, and the sum is over all fermions, not only the electrons and positrons, as shown in that equation. $G_{\mu\nu}^i$, $F_{\mu\nu}^B$ and the operators $\not{D}$ and $D$ are defined as

$$G_{\mu\nu}^j = \partial_\mu W_\nu^j - \partial_\nu W_\mu^j - g\epsilon^{jkl}W_\mu^k W_\nu^l \tag{8}$$

$$F_{\mu\nu}^B = \partial_\mu B_\nu - \partial_\nu B_\mu \tag{9}$$

$$\not{D}\Psi_{L,R}^{(j)} \equiv \gamma^\mu(\partial_\mu + igW_\mu + ig'Y_{L,R}B_\mu)\Psi_{L,R}^{(j)} \tag{10}$$

$$D \equiv (\partial_\mu + igT^j W_\mu^j + ig'YB_\mu) \tag{11}$$

where Greek indices in the subscript run from 0 to 3. The superscripts $j, k, l$ in Equations (7), (8), and (11) can take values from 1 to 3. $g$ and $g'$ are coupling constants, $\gamma^\mu$ are the gamma matrices, and $W_\mu$ and $B_\mu$ are two gauge bosons. $T^i$ is the generator of $SU(2)_L$, which is also a form of Pauli matrices, and $Y$ is the hypercharge generator of the $U(1)$.

The CP-conserving real type-I 2HDM potential of Equation (3) can be expressed as:

$$V(\Phi_1, \Phi_2, T) = V_{tree}(\Phi_1, \Phi_2) + V_{CW}(\Phi_1, \Phi_2) + V_T(T) + V_{\text{daisy}}(T), \tag{12}$$

where the terms are the tree level potential, the Coleman–Weinberg, the temperature corrections, and the daisy term, respectively. The individual terms are described in the following equations.

The tree level potential is given by:

$$\begin{aligned}
V_{\text{tree}}(\Phi_1, \Phi_2) =& m_{11}^2 \Phi_1^\dagger \Phi_1 + m_{22}^2 \Phi_2^\dagger \Phi_2 - \left[m_{12}^2 \Phi_1^\dagger \Phi_2 + m_{12}^* \Phi_2^\dagger \Phi_1\right] + \frac{1}{2}\lambda_1\left(\Phi_1^\dagger \Phi_1\right)^2 \\
&+ \frac{1}{2}\lambda_2\left(\Phi_2^\dagger \Phi_2\right)^2 + \lambda_3\left(\Phi_1^\dagger \Phi_1\right)\left(\Phi_2^\dagger \Phi_2\right) + \lambda_4\left(\Phi_1^\dagger \Phi_2\right)\left(\Phi_2^\dagger \Phi_1\right) \\
&+ \left[\frac{1}{2}\lambda_5\left(\Phi_1^\dagger \Phi_2\right)^2 + \frac{1}{2}\lambda_5^*\left(\Phi_2^\dagger \Phi_1\right)^2\right].
\end{aligned} \tag{13}$$

Here, $m_{12}$ depends on the mixing angle $\beta$ between $v1$ and $v2$ and is defined as $tan\,\beta = v2/v1$. While $tan\,\beta$ is a physical parameter here, it is still possible to redefine the two doublets and go to a basis of the two doublets: the so-called Higgs basis (H1 and H2)—for more detail, see [16]. The $\lambda$s are the quartic coupling constants, and all of them (including $\lambda_5$) are assumed real for this work and are given by:

$$\lambda_1 = \frac{1}{v_{\text{sm}}^2 \cos^2 \beta}\left(-\mu^2 \tan\beta + m_h^2 \sin^2\alpha + m_H^2 \cos^2\alpha\right) \tag{14}$$

$$\lambda_2 = \frac{1}{v_{\text{sm}}^2 \sin^2 \beta}\left(-\frac{\mu^2}{\tan\beta} + m_h^2 \cos^2\alpha + m_H^2 \sin^2\alpha\right) \tag{15}$$

$$\lambda_3 = \frac{1}{v_{\text{sm}}^2}\left(-\frac{2\mu^2}{\sin 2\beta} + 2m_{H_\pm}^2 + \left(m_H^2 - m_h^2\right)\frac{\sin 2\alpha}{\sin 2\beta}\right) \tag{16}$$

$$\lambda_4 = \frac{1}{v_{\text{sm}}^2}\left(\frac{2\mu^2}{\sin 2\beta} + m_A^2 - 2m_{H_\pm}^2\right) \tag{17}$$

$$\lambda_5 = \frac{1}{v_{\text{sm}}^2}\left(\frac{2\mu^2}{\sin 2\beta} - m_A^2\right) \tag{18}$$

The mass of $m_h, m_{H_\pm}, m_H, m_A$, and of the other particles depends on the location of the second minimum, i.e., on $(v_1, v_2)$. Here, $\alpha$ and $\beta$ are the mixing angles, and $v_{\text{sm}}$ is the vacuum expectation value in the SM. The details about scanning of the parameter space can be found in the recent works [17,18].

The other correction terms of the potential are defined as follows:

$$V_{\text{CW}}(v_1 + v_2) = \sum_j \frac{n_j}{64\pi^2}(-1)^{2s_j} m_j^4(v_1, v_2)\left[\log\left(\frac{m_j^2(v_1, v_2)}{\mu^2}\right) - c_j\right] \tag{19}$$

$$V_T = \frac{T^4}{2\pi^2}\left(\sum_{j=\text{bosons}} n_j J_B\left[\frac{m_j^2(v_1, v_2)}{T^2}\right] + \sum_{j=\text{fermions}} n_j J_F\left[\frac{m_j^2(v_1, v_2)}{T^2}\right]\right) \tag{20}$$

where $\mu$ is the renormalisation scale, which we take to be 246 GeV. As mentioned earlier, all the masses $m_j(v_1, v_2)$, $n_j$, and $s_j$ are mentioned in Appendix B of [19].

$J_B$ and $J_F$ are approximated in the Landau gauge up to the leading orders:

$$T^4 J_B\left[\frac{m^2}{T}\right] = -\frac{\pi^4 T^4}{45} + \frac{\pi^2}{12}T^2 m^2 - \frac{\pi}{6}T(m^2)^{3/2} - \frac{1}{32}m^4 \ln\frac{m^2}{a_b T^2} + \cdots, \tag{21}$$

$$T^4 J_F\left[\frac{m^2}{T}\right] = \frac{7\pi^4 T^4}{360} - \frac{\pi^2}{24}T^2 m^2 - \frac{1}{32}m^4 \ln\frac{m^2}{a_f T^2} + \cdots, \tag{22}$$

where $a_b = 16a_f = 16\pi^2 \exp(3/2 - 2\gamma_E)$ with $\gamma_E$ being the Euler–Mascheroni constant.

The daisy correction term is given by [20]

$$V_{\text{daisy}}(T) = -\frac{T}{12\pi}\left[\sum_{j=1}^{n_{\text{Higgs}}}\left((\bar{m}_j^2)^{3/2} - (m_j^2)^{3/2}\right) + \sum_{j=1}^{n_{\text{gauge}}}\left((\bar{m}_j^2)^{3/2} - (m_j^2)^{3/2}\right)\right] \tag{23}$$

As the temperature drops down below the critical temperature $T_c$, a second local minimum at $(\Phi_1 = v_1, \Phi_2 = v_2)$ appears with the same height of the global minimum situated at $\langle\Phi_1\rangle = \langle\Phi_2\rangle = 0$, and $T_c$ is determined by

$$V_{\text{tot}}(\Phi_1 = 0, \Phi_2 = 0, T_c) = V_{\text{tot}}(\Phi_1 = v_1, \Phi_2 = v_2, T_c). \tag{24}$$

We assumed that dark matter and other components might have been present but they did not contribute much to the energy density of the universe during the particular epoch of EWPT, which happened during radiation domination.

The early universe was flat, and hence the metric $g_{\mu\nu} = (+, -, -, -)$. The energy density of the homogeneous classical field $\Phi$:

$$\rho = \partial^0\Phi_a^\dagger\partial^0\Phi_a - (\mathcal{W}^0\Phi_a)^\dagger\mathcal{W}^0\Phi_a - (\mathcal{W}^j\Phi_a)^\dagger\mathcal{W}_j\Phi_a$$
$$+ \left[V_{\text{tot}}(\Phi_1, \Phi_2, T) - \mathcal{L}_{\text{gauge,kin}} - \mathcal{L}_f - \mathcal{L}_{\text{Yuk}}\right] \tag{25}$$

where, due to the condition of homogeneity and isotropy, all the spatial derivatives of Higgs fields are zero. Similarly,

$$\rho + P = 2\partial^0\Phi_a\partial^0\Phi_a^\dagger - i(\mathcal{W}^0\Phi_a)^\dagger(\partial_0\Phi_a) + i(\partial^0\Phi_a^\dagger)\mathcal{W}_0\Phi_a. \tag{26}$$

The Higgs fields start to oscillate around the second minimum, the minimum that appeared during the EWPT. Particle production from this oscillating field causes the

damping. The characteristic time of decay is equal to the decay width of the Higgs bosons. If this is large in comparison to the expansion and, thus, the universe cooling rate. Then, we may assume that Higgs bosons essentially live in the minimum of the potential. This was clearly discussed in [12].

Following the above assumption to be valid,

$$\rho = \dot{\Phi}_{a,\min}^2 + V_{\text{tot}}(\Phi_1, \Phi_2, T) + \frac{g_* \pi^2}{30} T^4. \tag{27}$$

The last term in Equation (27) arises from the Yukawa interaction between fermions and Higgs bosons and from the energy density of the fermions, the gauge bosons, and the interaction between the Higgs and gauge bosons. This is the energy density of the relativistic particles, which have not gained mass until the moment of EWPT. Essentially, the energy density of the plasma consists of two parts—namely, the energy density of the fields at the minima and the relativistic matter sector.

To calculate the entropy production, it is necessary to solve the evolution equation for energy density conservation,

$$\dot{\rho} = -3\mathcal{H}(\rho + P). \tag{28}$$

Henceforward, computational analysis is used for the calculations, which are discussed in the next section.

## 3. Numerical Calculations

In this section, we provide details regarding the numerical calculations, which were conducted. In passing, we discuss the C++ programming package that we used for the calculations.

BSMPT [20–22] is a C++ tool used for calculating the strength of the electroweak phase transition in extended Higgs sectors. This is based on the loop-corrected effective potential, also including daisy re-summation of the bosonic masses. The program calculates the vacuum expectation values of the potential (VEV) ($v$) as a function of temperature and, in particular, its value at the critical temperature $T_c$. The models implemented within this tool include CP-conserving 2HDM and next-to-minimal 2HDM. In this work, we restrict ourselves to the real sector of CP-conserving 2HDM potential.

This tool was used to calculate the VEVs, $T_c$s, and the effective value of potential $V_{eff}(T)$s for each benchmark point in Table 1. The parameters were chosen to satisfy the limiting conditions for type-I real 2HDM. We assumed that $VEV/Tc > 0.02$. Only those benchmark points are used that satisfy the conditions for first order phase transition. For details regarding the benchmark points and parameter space used here, please see [23]. The differential equation in Equation (28) was solved numerically for all the benchmark points in Table 1, and the entropy production was calculated for each of them as shown in Table 1. Some selected figures showing this production are given in Figure 1.

It is clear from Figure 1 that, as the critical temperature increases, the entropy released into the primeval plasma also increases correspondingly. All these productions were higher than the entropy production in the SM, which was about 13%. The main contribution for 2HDM scenarios came from the massive scalar bosons, which were absent in the SM.

**Table 1.** 2HDM Benchmark points for entropy production.

| | $m_h$ [GeV] | $m_H$ [GeV] | $m_{H^\pm}$ [GeV] | $m_A$ [GeV] | $\tan\beta$ | $\cos(\beta-\alpha)$ | $m_{12}^2$ GeV$^2$ | $\lambda_1$ | $\lambda_2$ | $\lambda_3$ | $\lambda_4$ | $\lambda_5$ | $T_c$ | $vev/T_c$ | $\delta s/s$ [%] |
|---|---|---|---|---|---|---|---|---|---|---|---|---|---|---|---|
| BM1 | 125 | 500 | 500 | 500 | 2 | 0 | $10^5$ | 0.258 | 0.258 | 0.258 | 0 | 0 | 161.36 | 1.4 | 57 |
| BM2 | " [1] | " | " | " | " | 0.06 | " | 1.14 | 0.037 | 0.63 | 0 | 0 | 167.95 | 1.6 | 59 |
| BM3 | " | " | " | " | 10 | 0 | 24,752.5 | 0.258 | 0.258 | 0.258 | 0 | 0 | 161.02 | 1.4 | 56 |
| BM4 | " | " | " | " | 10 | 0.1 | 24,752.5 | 4.13 | 0.22 | 4.15 | 0 | 0 | 255.71 | 1.9 | 73 |
| BM5 | " | " | " | 485 | 2 | 0.0 | $10^5$ | 0.26 | 0.26 | 0.26 | −0.244 | 0.244 | 161.53 | 1.4 | 57 |
| BM6 | " | " | " | 485 | 2 | 0.07 | $10^5$ | 1.28 | 0.002 | 0.7 | −0.244 | 0.244 | 169.81 | 1.7 | 60 |
| BM7 | " | " | " | 477 | 2 | 0.07 | $10^5$ | 1.28 | 0.002 | 0.7 | −0.37 | 0.37 | 169.53 | 1.7 | 60 |
| BM8 | " | " | " | 485 | 10 | 0.0 | 24,752.5 | 0.258 | 0.258 | 0.258 | −0.244 | 0.244 | 160.76 | 1.3 | 56 |

**Table 1.** *Cont.*

| | $m_h$ [GeV] | $m_H$ [GeV] | $m_{H\pm}$ [GeV] | $m_A$ [GeV] | $\tan \beta$ | $\cos(\beta - \alpha)$ | $m_{12}^2$ GeV$^2$ | $\lambda_1$ | $\lambda_2$ | $\lambda_3$ | $\lambda_4$ | $\lambda_5$ | $T_c$ | $vev/T_c$ | $\delta s/s$ [%] |
|---|---|---|---|---|---|---|---|---|---|---|---|---|---|---|---|
| BM9 | " | " | " | 350 | 10 | 0.1 | 24,752.5 | 4.13 | 0.22 | 4.15 | −2.1 | 2.1 | 209.87 | 1.8 | 68 |
| BM10 | " | " | 485 | 500 | 2 | 0.00 | $10^5$ | 0.258 | .258 | −0.23 | 0.49 | 0 | 153.27 | 1.25 | 53 |
| BM11 | " | " | 485 | 500 | 2 | 0.07 | $10^5$ | 1.28 | 0.002 | 0.21 | 0.49 | 0 | 169.28 | 1.7 | 60 |
| BM12 | " | " | 485 | 500 | 10 | 0.0 | 24,752.5 | 0.258 | 0.258 | −0.23 | 0.49 | 0 | 160.51 | 1.3 | 56 |
| BM13 | " | " | 485 | 500 | 10 | 0.1 | 24,752.5 | 4.13 | 0.22 | 3.66 | 0.49 | 0 | 241.75 | 1.88 | 70 |
| BM14 | " | " | 485 | 485 | 2 | 0 | $10^5$ | 0.258 | 0.258 | −0.23 | 0.24 | 0.24 | 159.76 | 1.3 | 59 |
| BM15 | " | " | 485 | 485 | 2 | 0.07 | $10^5$ | 1.28 | 0.002 | 0.21 | 0.244 | 0.244 | 168.61 | 1.7 | 59 |
| BM16 | " | " | 485 | 485 | 10 | 0 | 24,752.5 | 0.258 | 0.258 | −0.23 | 0.244 | 0.244 | 160.19 | 1.3 | 56 |
| BM17 | " | 485 | 485 | 485 | 2 | 0.0 | 94,090 | 0.258 | 0.258 | 0.258 | 0 | 0 | 161.31 | 1.4 | 57 |
| BM18 | " | 485 | 485 | 485 | 2 | 0.07 | 94,090 | 1.22 | 0.02 | 0.67 | 0 | 0 | 169.7 | 1.7 | 60 |
| BM19 | " | 485 | 485 | 485 | 10 | 0 | 23,289.6 | 0.258 | 0.258 | 0.258 | 0 | 0 | 160.96 | 1.3 | 57 |
| BM20 | " | 485 | 485 | 485 | 10 | 0.1 | 23,289.6 | 3.9 | 0.22 | 3.9 | 0 | 0 | 230.18 | 1.86 | 70 |
| BM21 | " | 485 | 485 | 500 | 2 | 0 | 94,090 | 0.258 | 0.258 | 0.258 | 0 | 0 | 161.31 | 1.4 | 57 |
| BM22 | " | 90 | 200 | 300 | 2 | 0 | 3240 | 0.258 | 0.258 | 1.31 | 0.3 | −1.35 | 150.76 | 1.2 | 51 |
| BM23 | " | 90 | 200 | 300 | 10 | 0 | 801.98 | 0.258 | 0.258 | 1.31 | 0.3 | −1.35 | 135.38 | 1.06 | 37 |
| BM24 | " | 90 | 200 | 300 | 10 | 0.2 | 801.98 | 0.263 | 0.258 | 1.06 | 0.3 | −1.35 | 141.06 | 1.1 | 42 |

[1] It means the present value follows the previous value in the same column.

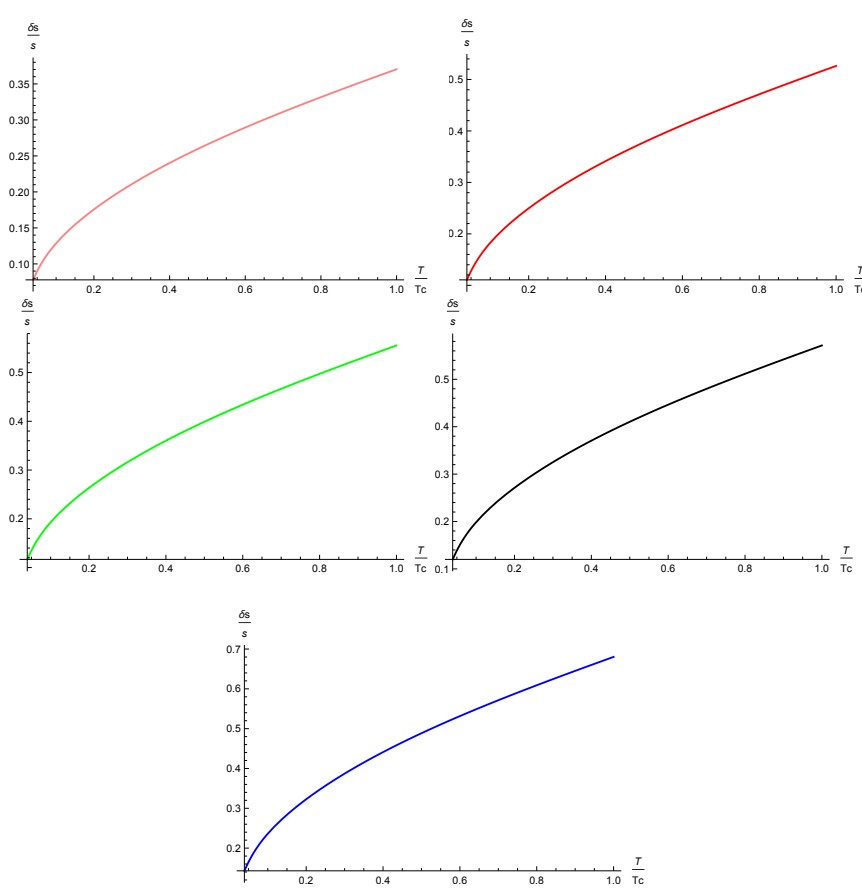

**Figure 1.** The figures show the entropy production for five different benchmark (BM) points: Pink line (BM 23, $T_c = 135.38$ GeV and $\delta s/s = 37\%$), Red line (BM 10, $T_c = 153.27$ GeV and $\delta s/s = 53\%$), Green line (BM 1, $T_c = 161.36$ GeV and $\delta s/s = 30\%$), Black line (BM 15, $T_c = 168.61$ GeV and $\delta s/s = 59\%$), and Blue line (BM 20 $T_c = 230.18$ GeV and $\delta s/s = 70\%$).

## 4. Conclusions

We demonstrated in this paper that, even in the minimal extension of SM with two Higgs bosons, EWPT is of the first order, and the net production of entropy is very large

compared to the SM. An interesting point to be noted is that $g_\star$ decreases as the universe cools down. However, as the temperature approaches the decoupling temperatures of electrons, the contribution to the entropy becomes similar to that of SM.

Each and every benchmark point is calculated using BSMPT with the limiting condition $VEV/T_c > 0.02$. This condition can be modified and it can become even smaller, which can give rise to more benchmark points. Overall, for the case of real Type-I 2HDM, the effect will be similar, and it will not vary. However, this effect will change if other variants of 2HDM are considered.

This influx of entropy into the primeval plasma has many implications in cosmology. First, it can dilute the preexisting number density of frozen out spices, for example dark matter. The freezing out of dark matter can happen before or after EWPT depending on the models and their parameters. The number density of decoupled dark matter is measured relative to that of CMB photons. More entropy production increases in the density of photons can dilute the preexisting decoupled dark matter number density. Thus, models describing cold dark matter require adjustment accordingly. Secondly, if there exists a preexisting baryon asymmetry, this influx can highly dilute it.

Not only this process but also phase transition at the QCD epoch or the evaporation of PBHs produce entropy and can dilute pre-existing baryon asymmetry and dark matter density [24]. The extended standard model processes are under observation at the present time.

**Author Contributions:** Article by A.C., M.Y.K. and S.P.; A.C. and M.Y.K. contributed equally to this work; and S.P. helped to prepare the manuscript. All authors have read and agreed to the published version of the manuscript.

**Funding:** The work of A.C. and S.P. were funded by RSCF Grant 19-42-02004. The research by M.Y.K. was financially supported by a grant of the Russian Science Foundation (Project No-18-12-00213-P).

**Institutional Review Board Statement:** Not applicable.

**Informed Consent Statement:** Not applicable.

**Data Availability Statement:** Not applicable.

**Conflicts of Interest:** There are no conflict of interest among the authors.

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
