# Peer review of "Effects of 2HDM in Electroweak Phase Transition"

_galaxies, doi:10.3390/galaxies9020045_

Round 1
Reviewer 1 Report
In this work, the authors discuss the possibility of first-order phase transition in the type-I 2HDM and estimate the entropy production at some benchmark points. The discussion of first-order phase transition is important, because it is related to the physics beyond the SM. Most of my questions and suggestions are related to the phase transition.
Questions/Suggestions:
- The authors discuss the muon g-2 in the first part of Introduction, but it is not discussed in main topic. I recommend to write more detail of first-order phase transition or entropy production in this part. For example, why entropy production is important, how to realize first-order phase transition or not.
- The authors choose the example of the theory beyond the SM as supersymmetric model. However, the authors discuss the first-order phase transition in Type-I 2HDM, although minimum supersymmetric model is related to Type-II. In this paper, the other types of 2HDM is not much discussed. I think, it is discussed by "But this effect will change if other variants of 2HDM are considered" in Summary. But, I recommend to discuss the detail of the types in content of this paper, for example, the definition of Z_2 charge, what is difference among types...
- I recommend to write the daisy re-summation effects in section 2. Maybe the C++ programming can automatically calculate these effects, however, you should add the formula of them below Eq.(22).
- From this paper, I cannot understand why the authors choose the benchmark points in Table 1. I recommend to write the experimental and theoretical constraints in this model before discussing the results with benchmark points.
- There are some papers wherein the possibility of first-order phase transition in 2HDM, such as arXiv:1612.04086 [hep-ph]. Therefore I recommend to cite the papers about that and to discuss the different of analysis among this works and them.
- The analysis shows that first-order electroweak phase transitions can be achieved, but does not specify their strength. Therefore I recommend to add the value of VEV/T_c into Table 1.
- In this model, there are some possibilities of path of phase transition. In benchmark points in Table 1, how is the path of phase transition? (0,0) \to (v_1, v_2)? or (0,0) \to (0, v_2') \to (v_1, v_2)? Does your results depends on the path of phase transition?
From them, I recommend a drastic rewrite.
Author Response
Texts in the introduction and conclusion are added. One term in equation (12) and a new equation (23) are added. Table 1 is modified.

Reviewer 2 Report
My comments are as follows:
1) Under Eq.(24), the authors said "due to the condition of homogeneity and isotropy, all the spatial derivatives of Higgs fields are zero", I would suggest author to make sure about that, since these part would be important for the generation of gravitational wave during bubbles collision stage;
2) The importance of m_12^2 for the EWPT should be added ( see the paper published in JHEP05(2018)151 );
3) Discussions on cosmological implication of entropy production during EWPT can be added.
Author Response
We thank the reviewer for his comments. Texts in the introduction and conclusion are added. One term in equation (12) and a new equation (23) are added. Table 1 is modified.

Reviewer 3 Report
The two-Higgs-doublet model (2HDM) is used by authors of this paper for revitalization the entropy production scenarios in the electroweak phase transition. It is stressed that this revitalization is motivated in part by the experimental anomalous value of the muon magnetic moment (g-2 experiment at FERMILAB) which is incompatible with the Standard Model. Authors investigate in details the specific two Higgs doublet model(2HDM) for describing the Electroweak Phase Transition (EWPT) in the early Universe. In particular, authors provide the detailed theoretical description of the Electroweak Phase Transition in the two-Higgs-doublet model. The major result of this paper is the numerical calculation of the entropy production during Electroweak Phase Transition. It is demonstrated that the net production of entropy is much more higher than in the Standard Model. All these results are new and physically interesting. This paper, in spite of some formal presentation, may be interesting for a wide scientific community in view of the fundamental importance in understanding the properties of the early Universe and the possible extensions of the Standard Model. This manuscript is well organized, clearly written.
Author Response

(The authors gave the same response as above.)

Round 2
Reviewer 1 Report
I think that this revised paper is sufficiently improved.